



# OzRiCa: An Australian riverine carbon database of concentrations, gas fluxes and isotopes

Francesco Ulloa-Cedamanos[1], Adam T. Rexroade[1], Yihan Li[1], Lindsay B. Hutley[1], Wei Wen Wong[2], Marcus B. Wallin[3], Josep G. Canadell[4], Anna Lintern[5], Clément Duvert[1,6]

[1]Research Institute for the Environment and Livelihoods, Charles Darwin University, Darwin, NT, Australia
[2]Water Studies, School of Chemistry, Monash University, Melbourne, VIC, Australia
[3]Department of Aquatic Sciences and Assessment, Swedish University of Agricultural Sciences, Uppsala, Sweden
[4]CSIRO Environment, Canberra, ACT, Australia
[5]Department of Civil Engineering, Monash University, Clayton, VIC, Australia
[6]College of Science and Engineering, James Cook University, Cairns, QLD, Australia

*Correspondence to*: Francesco Ulloa-Cedamanos (francesco.ulloacedamanos@cdu.edu.au); Clément Duvert (clem.duvert@cdu.edu.au)

**Abstract.**

Understanding carbon (C) cycling in riverine ecosystems is crucial for accurate estimates of regional and global C budgets. However, the slow progress in identifying spatial and temporal patterns and drivers of riverine C has been largely driven by limited data availability. This lack of information is particularly acute in Australia. To address this issue, we compile the first comprehensive database (OzRiCa) of concentrations, stable isotopes, and fluxes of particulate organic C (POC), dissolved organic (DOC) and inorganic C (DIC), carbon dioxide ($CO_2$) and methane ($CH_4$) for streams and rivers across Australia, along with concurrent physical and chemical data. The OzRiCa database contains 54,843 observations from 2,879 unique sites derived from state agencies, scientific publications, and newly collected, previously unpublished data. The measurements span the period between 1966 to 2024. The database reveals that most observations and sites are located in the Mediterranean, temperate and subtropical regions of the country, with fewer data in semi-arid and tropical regions. We also highlight patterns in temporal data coverage, with half of the sites limited to a single measurement. The scarcity of long-term and high-resolution temporal data for $CO_2$ and $CH_4$, along with the limited number of direct measurements for DIC and $CO_2$, emphasises the need for more direct, frequent and long-term monitoring efforts to capture the extreme temporal variability of Australian climatic regions. We also identify limited stable isotopic data and concurrent measurement of multiple C species, both limiting our ability to better understand C sources and sinks, as well as in-stream C processes. Future research should prioritise these critical gaps to improve our understanding of riverine C dynamics in Australia. OzRiCa provides a baseline for future research, paving the way for studies of fluvial C fluxes at regional, national, and global scales. This database will also be a valuable resource to studying ecosystem health, water quality, and other biogeochemical processes.

Keywords. Carbon cycling; Lotic ecosystems; Aquatic fluxes; Monitoring; Dynamics; Variability.

## 1. Introduction

Carbon (C) is a fundamental element that continuously cycles between the atmosphere, ocean, and terrestrial stores. A large portion of the C fixed by terrestrial ecosystems, along with the C derived from rock weathering, is transferred laterally via C-rich groundwater and soil water into inland waters (Abril and Borges, 2019; Regnier et al., 2022). Once in running waters, C can be transported downstream, released back into the atmosphere, transformed within the water column, or stored in sediments (Bauer et al., 2013; Cole et al., 2007). Understanding

these pathways is critical for better estimating the global C budget (Regnier et al., 2022), predicting and mitigating future C dynamics (Friedlingstein et al., 2023) and climate change (Lauerwald et al., 2020; Regnier et al., 2022). Lateral C transfer and processing shapes water quality and in turn ecosystem health of inland waters (Stewart et al., 2024), drives stream biodiversity, food web structures (O'Donnell et al., 2020; Xenopoulos et al., 2021), water clarity (Fabricius et al., 2014), pH and buffering capacity (Cai et al., 2011), the mobility of toxic metals (Laudon

et al., 2012), and connects with other biogeochemical elemental cycles (Moran et al., 2016).

Within inland waters, C exists primarily in dissolved and particulate forms. Dissolved C includes two components. First, non-gas forms, such as inorganic (DIC) and organic C (DOC), which have been the focus of numerous studies and reviews (e.g. Bakalowicz, 1975; Hope et al., 1994; Schlesinger and Melack, 1981; Ulloa-Cedamanos et al., 2020). Second, gaseous forms such as carbon dioxide ($CO_2$) and methane ($CH_4$), which have

garnered increasing attention due to their role as greenhouse gases (GHG) (Lauerwald et al., 2023), with inland waters recognised as hotspots for these emissions to the atmosphere (IPCC, 2013). Particulate C, primarily in the form of particulate organic C (POC) remains the least constrained form of inland water C fluxes, particularly in small mountainous catchments (Liu et al., 2024; Xu et al., 2021). Carbon in inland water systems can originate from either biogenic or geogenic sources. The former derives from both the decomposition of reactive organic

matter produced by terrestrial and aquatic vegetation (Stockmann et al., 2013) and autotrophic and heterotrophic respiration (Li, 2019), while the latter originates from rock and mineral weathering (Amiotte Suchet and Probst, 1993; Berner et al., 1983). Stable C isotopes have proven valuable for tracing these sources (Campeau et al., 2017; Ulloa-Cedamanos et al., 2021; Wang et al., 2023).

Despite substantial advancements in statistical and process-based models, upscaling models of inland water

C fluxes often lead to poorly constrained estimates due to the limited availability and quality of C observations (Deemer et al., 2016; Liu et al., 2024; Regnier et al., 2013). This is reflected in the range of estimated inland water GHG fluxes in the sixth assessment report (AR6), which ranged from 0.8-1.2 Pg C $yr^{-1}$ for $CO_2$ and 117-212 Tg C $yr^{-1}$ for $CH_4$ (Canadell et al., 2023). A major issue contributing to the large uncertainties and low confidence of these C fluxes is the inherent spatial and temporal variability of C species in riverine waters, coupled with limited

coverage of site-specific C observations. For instance, tropical regions are recognised as hotspots for C cycling yet are underrepresented in global databases (Lauerwald et al., 2023). These regions experience highly variable flow regimes and extreme weather events but have limited temporal data coverage. Similarly, small headwater streams, which are typically not monitored (Marx et al., 2017), likely contribute disproportionally to total C emission fluxes (Raymond et al., 2013). Moreover, most studies tend to focus on one or two C species, yet multiple

C species should be collected concurrently to provide a complete C balance (Vachon et al., 2021). These knowledge gaps lead to large uncertainties when upscaling (Friedlingstein et al., 2023; Villalobos et al., 2023), highlighting the need for comprehensive data collection across different inland water bodies in contrasting climatic regions (Liu et al., 2024). To address these uncertainties, increasing inland water C measurements is



necessary, as is the need to compile such data into well-curated and openly accessible databases. Streams and rivers are systems that require particular attention given their overwhelming importance in global inland water C export (Lauerwald et al., 2023). Recent efforts have been directed towards developing global riverine databases for DIC and DOC (Hartmann et al., 2014; Liu et al., 2024; Virro et al., 2021), $CO_2$ (Gómez-Gener et al., 2021; Liu et al., 2022c), and $CH_4$ (Stanley et al., 2023) concentrations and fluxes. Similarly, regional initiatives have compiled riverine C data for the Amazon River (DIC and DOC, Mayorga et al., 2012), the Congo River (DOC, Hemingway et al., 2017), and major rivers in the US ($CO_2$, Butman and Raymond, 2011; Jones Jr. et al., 2003), China ($CO_2$, Dong Liu et al., 2022; Ran et al., 2021), and Africa ($CO_2$ and $CH_4$, Borges et al., 2015).

Australia, with its wide range of climatic regions, has a large variability in river chemistry across its heterogeneous landscapes (Liu et al., 2022b). A recent global analysis of dissolved C export from rivers identified Australia as the region with largest discrepancies across models (Liu et al., 2024), likely due to the limited spatial and temporal representativity and low availability of C observations from streams and rivers. While numerous studies have focused on C cycling in lakes, reservoirs and farm dams across Australia (e.g. Bastien and Demarty, 2013; Grinham et al., 2018; Malerba et al., 2024; Sturm et al., 2014), riverine systems have received comparatively less attention despite their extensive coverage. However, in recent years a number of local-scale studies have begun to explore riverine C dynamics in tropical (e.g. Duvert et al., 2020a; Rosentreter and Eyre, 2020; Solano et al., 2024), subtropical (e.g. Andrews et al., 2021; Jeffrey et al., 2018), temperate (e.g. Hancock et al., 2022), Mediterranean (e.g. Nelson et al., 1996; Kostoglidis et al., 2005), and semi-arid and arid (e.g. Bargrizan et al., 2022; Biswas and Mosley, 2019) streams and rivers of the Australian continent. In addition, state agencies have been monitoring various C species, including alkalinity, DIC and DOC, in rivers across the country since the 1960s. Despite the growing availability of riverine C data, a recent effort to quantify the national C budget for Australia, that included an estimate of lateral C fluxes, resulted in large uncertainties in this flux due to reliance on data sourced from global databases and modelling, rather than data specific to Australia (Villalobos et al., 2023). Hence, there is a clear need for a coherent database that aggregates published data from studies across Australia.

The OzRiCa (Australian Riverine Carbon) database is the first effort to compile all existing riverine C data from state agencies, the scientific literature and newly collected and previously unpublished data in Australia. This resource is designed not only to help in reducing uncertainties in future C budget estimations across scales, but also in assessing ecosystem health, water quality and other biogeochemical cycles linked to C cycling. In this study, we (1) provide a detailed description of the collection, processing, and construction of OzRiCa, (2) summarise key spatial and temporal patterns of C concentrations, fluxes, and stable isotopes in the database, and (3) identify critical data gaps and future opportunities to improve our current understanding of riverine C dynamics in Australia.



## 2. Methods

### 2.1. Data extraction from the literature

We conducted a Web of Science search in January 2024 using the following Boolean query within the 'Topic' field: *Australia AND ("dissolved organic carbon" OR "dissolved inorganic carbon" OR DIC OR DOC OR $CO_2$ OR $CH_4$ OR methane OR "carbon dioxide") AND (river OR stream OR creek OR freshwater)*. We excluded review articles and papers outside the scope of this study, such as papers based on data from brackish environments (e.g. estuaries). For studies involving transect sampling from river to estuary, we only included data from the freshwater endmember. We also included data from theses that did not result in scientific publications. In total, we extracted data from 64 papers and 2 theses.

For each paper, we mined all available data, including geographic coordinates, catchment area, discharge, stream order, water physico-chemical parameters (temperature, pH, conductivity, dissolved oxygen), sampling date and time, $CO_2$ concentration, $CO_2$ measurement method (whether direct or indirect), $CH_4$ concentration, $CO_2$ flux, $CH_4$ flux, DOC concentration, DIC concentration, POC concentration, alkalinity, and the isotopic ratio of carbon-13 to carbon-12 ($\delta^{13}C$) of different C species. We included $CO_2$ values derived indirectly from pH and alkalinity, as we assumed the authors had ensured their accuracy and reliability. When the required data were not available in the supporting information, we contacted the corresponding author(s) to request digital databases. As a last resort, we digitised data from plots using the WebPlotDigitizer tool (Rohatgi, 2024). Lastly, we standardised all C concentrations to $\mu mol\ L^{-1}$ and all $CO_2$ and $CH_4$ emission fluxes to $mmol\ m^{-2}\ d^{-1}$.

### 2.2. Data extraction from state agencies

A total of 35,776 coupled alkalinity, pH and temperature measurements and 16,853 DOC measurements from 2,287 sites were obtained from databases maintained by state departments. Data were obtained from five agencies: Department of Lands, Planning and Environment, Northern Territory (NT); Department of Environment, Science and Innovation, Queensland (QLD); WaterNSW, New South Wales (NSW); Department of Energy, Environment and Climate Action, Victoria (VIC); and Department of Water and Environmental Regulation, Western Australia (WA). These data are either in the public domain or published under the Creative Commons (CC-BY) license. We used the total alkalinity, water temperature and pH data to calculate DIC concentrations based on standard carbonate equilibrium equations (Kalka, 2021; Millero, 1995; Plummer and Busenberg, 1982). Indirect DIC calculations from alkalinity and pH are unlikely to be a source of significant uncertainty given the typical pH range of rivers (i.e. from 6 to 8). Unlike our approach for the literature-based dataset (section 2.1), we did not indirectly estimate $CO_2$ concentrations from alkalinity, pH and temperature for the state data. This decision was made to avoid potential errors, particularly in systems with low buffering capacity (e.g. headwater streams) and high DOC concentrations (e.g. forested streams and wetlands) (Abril et al., 2015; Liu et al., 2020), and because of the potentially lower reliability of pH data given the diversity of sources and methods across states.

### 2.3. Sampling campaigns

In addition to collecting and aggregating already published data, we collected new riverine C data from tropical and temperate streams across Australia. In February-April 2023 we sampled 97 sites during the wet season in the Australian tropics (NT and northern QLD). In September 2023, these same 97 sites were revisited and



sampled in the dry season for streams that had maintained flow. Sampling these sites twice was done to capture seasonal variability, with the first campaign occurring during the peak of the wet season (JFM) and the latter at the end of the dry season (JJAS). A third campaign was conducted in the temperate areas of VIC and Tasmania between November and December 2023, corresponding to the austral late spring and early summer. This third campaign increased the number of sampled sites by 68. All the sites were selected based on local knowledge, satellite imagery, and accessibility, with an emphasis on small headwater streams given the underrepresentation of low-order streams in national and global databases (Lauerwald et al., 2023; Liu et al., 2022c; Rocher-Ros et al., 2019).

At each site, samples for DIC, DOC, POC, $CO_2$ and $CH_4$ concentrations were collected and measured, although there were some exceptions to this due to resource constraints. In addition to these core C parameters, supplementary parameters were measured when appropriate and feasible. These included specific conductivity, pH, water temperature, dissolved oxygen, flow rate, flow velocity, fluxes of $CO_2$ and $CH_4$ to the atmosphere, gas transfer velocity, stream depth and width, and streambed slope (see Appendix A1).

The primary method employed for measuring DIC, $CO_2$, $CH_4$ concentrations, and their isotopic compositions involved collecting duplicate water samples in 12 mL Exetainer vials, preserved with 10 µL of 6% mercuric chloride ($HgCl_2$) and sealed with septa-caps, containing no headspace. Samples were analysed at the Water Studies Analytical and Stable Isotope Facility, Monash University (VIC). $CO_2$ and $CH_4$ concentrations were measured on a trace gas analyser (VICI TGA 6k; VICI Valco Instruments) interfaced with a pulse-discharge helium ioniser. Just prior to analysis, 4 mL of water was replaced with helium. Samples were shaken to allow gas exchange between the water and headspace, and the equilibrated headspace was then analysed for $CO_2$ and $CH_4$ on the trace gas analyser. DIC was converted to $CO_2$ by adding phosphoric acid (12.5 mM) to the samples and leaving them on a shaker table overnight. $CO_2$ was then analysed using the trace gas analyser. The isotopic signatures of DIC, $CO_2$ and $CH_4$ were determined using a Gas Bench II interfaced with a continuous-flow isotope ratio mass spectrometer (Thermo Scientific Delta V Advantage). For $\delta^{13}C$-$CH_4$ analysis, dissolved $CH_4$ in water samples was completely purged, extracted, and trapped using an automated trace gas cryo-prep system (Thermo Scientific PreCon) before being analysed on the isotope ratio mass spectrometer. DOC samples were collected in the field by filtering 40 mL of stream water through a 0.7 µm G/GF glass microfiber syringe filter (Whatman) into a pre-acidified (200 µL of 98% $H_2SO_4$) borosilicate amber vial. Samples were refrigerated and sent to the Environmental Analysis Laboratory at Southern Cross University (NSW) for analysis. DOC was measured using a total organic C analyser (TOC-VCPH; Shimadzu) and $\delta^{13}C$-DOC was measured using an O.I. Analytical 1010 TOC analyser interfaced to a Europa 20-20 isotope ratio mass spectrometer (PDZ Sercon Ltd.). To determine POC concentrations and isotopes, 1L water samples were filtered using a pre-weighed 0.7 µm G/GF glass microfiber filter, which was oven-dried at 450°C for 2 h, and weighed again. The filters were then analysed on an ANCA GSL2 elemental analyser interfaced to a Hydra 20-22 continuous-flow isotope ratio mass spectrometer (Sercon Ltd.) at Monash University.

Due to equipment availability at the sites in the NT and QLD, $CO_2$ and $CH_4$ concentrations were also measured using a headspace equilibration technique where 45 mL of water and 15 mL of air were equilibrated in a sealed 60 mL syringe by gently shaking for two minutes. The equilibrated gas was then transferred into a sealed 12 mL syringe via a three-way-stopcock and kept cool until analysed using a LI-7810 trace gas analyser (Li-Cor Inc.) within 12 hours. The original concentration of gas in the water was calculated using a mass balance of gas



in the equilibrated gas sample and the atmospheric gas used for mixing (Cawley et al., 2020). Despite sometimes measuring gas concentrations using both the headspace and Exetainer sampling methods, only one is reported in

the dataset. The selected method is specified in the database and details on how the chosen method was selected are in Appendix A2. This $CO_2$ concentration, in combination with water temperature and pH, was then used to calculate any missing DIC using carbonate equilibrium equations (Kalka, 2021; Millero, 1995; Plummer and Busenberg, 1982).

Fluxes of $CO_2$ and $CH_4$ to the atmosphere were measured using a tethered floating chamber connected to the

LI-7810 gas analyser. Flux was calculated using the slope of the linear increase of $CO_2$/$CH_4$ within the chamber. Linearity of concentration increase was assessed both visually and using simple linear regression to ensure the observed gas flux was solely diffusive flux and not ebullitive flux. Samples were removed from the dataset when the $R^2$ value was lower than 0.85. Two to three replicate chamber measurements, each lasting three to five minutes were undertaken at each reach to capture different stream morphologies (e.g. pools and riffles). The reported flux

was the weighted average of all replicates for the site, with weights assigned subjectively based on the distribution of the different hydro-morphological features that each replicate captured to limit the influence of extreme (high or low) fluxes when reporting only a mean value for the site. For sites where the stream was too turbulent to use a floating chamber, gas transfer velocity was estimated using propane gas injections (further details in Appendix A1).

### 2.4. OzRiCa database development and structure

OzRiCa comprises both existing published data (section 2.1 and section 2.2), and newly collected, unpublished data from our sampling campaigns as described above (section 2.3). The database consists of three tables (Tables A, B and C;Ulloa-Cedamanos et al., 2025). Table A contains the concentrations, fluxes and isotopic compositions of DIC, DOC, POC, $CO_2$ and $CH_4$ at all sites and for all sampling dates. While POC is included in

the database, it is not covered in the following sections due to the limited number of measurements across Australia (n=51). In Table A, sites with at least two records in different months within the same calendar year were categorised as "seasonal", whereas sites with at least two records across different calendar years were categorised as "interannual". The rest of sites were categorised as a "spot" measurement. Table B contains a range of catchment characteristics, including the climatic region and stream order of all sampled sites, which was obtained using the

Köppen-Geiger classification (Beck et al., 2018), and the National Environmental Stream Attributes database (National Environmental Stream Attributes v1.1.5. .dataset.) on ArcGIS software (ESRI, 2024). Table B also contains water quality parameters when available, including dissolved oxygen, specific conductivity, water temperature, and pH; geomorphological features of the site such as stream or river depth and width, flow rate and flow velocity, and catchment area. Table B is more comprehensive for sites derived from our sampling campaigns

than for those based on published data. Lastly, Table C lists the publications and datasets from which we extracted the data. A more detailed description of each table can be found in the corresponding metadata tables provided in Appendix B.

### 3. Data overview

3.1. Spatial data coverage

OzRiCa contains 54,843 observations from 2,879 unique sites across Australia (Figure 1). The distribution of these observations, however, varies significantly across hydrological and climatic regions. Most sites are concentrated in areas where streams and rivers are denser and less intermittent. Non-gas C observations (i.e. DIC and DOC; Figure 1A) are predominantly found in eastern (810 sites, 140°E-155°E) and western Australia (1,669 sites, 110°E-125°E), while gas C observations (i.e. $CO_2$ and $CH_4$; Figure 1B) are mostly concentrated along the

eastern coast and the Murray River in the southeast (285 sites).

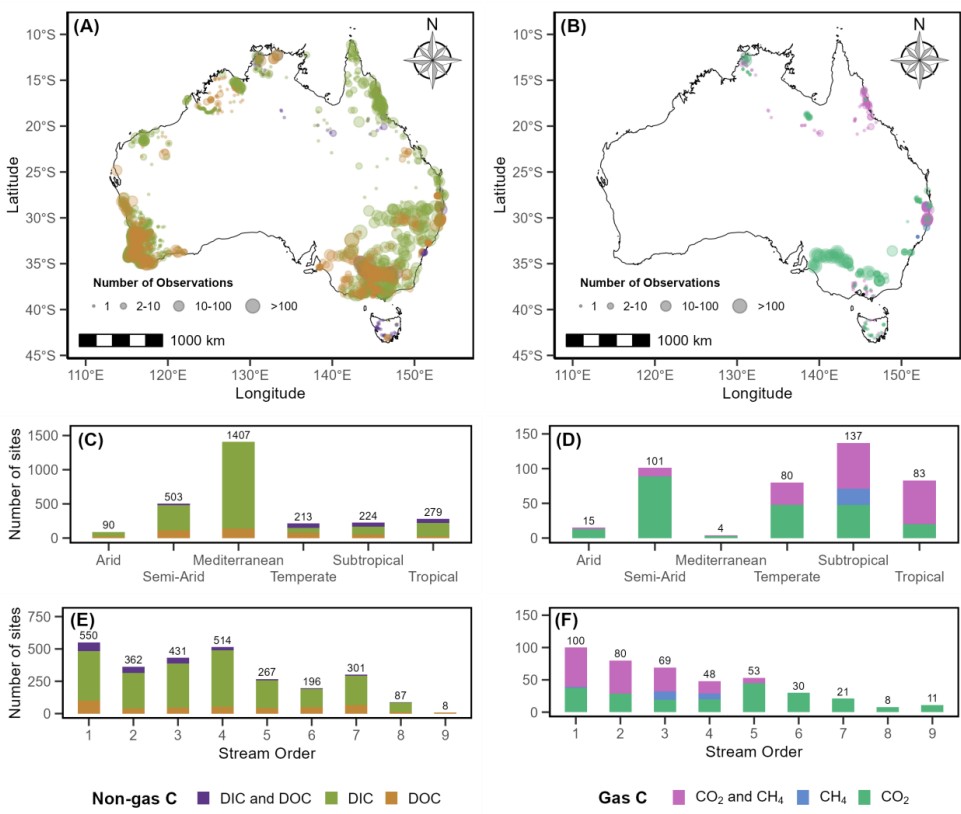

Figure 1. Spatial coverage of (A) non-gas C (DIC and DOC), and (B) gas C ($CO_2$ and $CH_4$) concentrations, with sites and C species grouped by (C, D) climate, and (E, F) stream order. Coloured points in panels A, B, C and D as per legend in panels E and F.

Although the arid climate zone dominates the Australian continent, covering over half of the country (Table 1), 97% of the data included in OzRiCa are from sites located in other climatic regions (Figure 1C and 1D). Site density is biased towards Mediterranean, subtropical and temperate regions, with over 750 sites per $10^6$ $km^2$ of terrestrial surface area (Table 1). While we cannot entirely rule out the possibility of site selection bias, the higher number of sites in Mediterranean, subtropical and temperate regions is expected, as subtropical and temperate





regions have higher river density and flow permanence, and the Mediterranean region has rich data coverage due to long-term C monitoring across southwestern WA since 1969. At the other end of the spectrum, sites in arid regions exhibited the lowest representation, as river systems in this climate zone are very limited in number, sparse, and often highly intermittent.

Table 1. Sites, C observations and temporal distribution across different climatic regions in OzRiCa. N refers to the number of
observations; non-gas C refers to DIC and DOC, and gas C refers to $CO_2$ and $CH_4$.

| Climate | Fractional area | Number of Sites | Site density | Non-gas C (N=2744) | Gas C (N=420) | Spot (N=1410) | Seasonal (N=372) | Interannual (N=1095) |
|---|---|---|---|---|---|---|---|---|
| | % | N | sites/($10^6$ km²) | % of sites | % of sites | % of sites | % of sites | % of sites |
| Arid | 53 | 96 | 25 | 3 | 4 | 3 | 1 | 5 |
| Semi-Arid | 27 | 589 | 283 | 19 | 24 | 27 | 7 | 17 |
| Mediterranean | 3 | 1410 | 6775 | 51 | 1 | 45 | 54 | 52 |
| Temperate | 4 | 222 | 766 | 8 | 19 | 7 | 13 | 7 |
| Subtropical | 5 | 279 | 773 | 8 | 33 | 9 | 13 | 10 |
| Tropical | 9 | 283 | 412 | 10 | 20 | 10 | 12 | 8 |

The distribution of sites in the database was also evaluated in terms of system size, using stream order as a proxy for size (Figure 1E and 1F). Here the distribution was skewed towards sites draining small catchments (stream orders 1-3), accounting for 1,344 sites out of 2,744 sites for non-gas C data and 249 sites out of 420 for gas C data. Within these low-order streams, 11% (non-gas C) and 61% (gas C) of the data are unpublished as they
originate from our field campaigns.

DIC and DOC were the most prevalent C species across all climatic regions (Figure 1C and 1D) and stream orders (Figure 1E and 1F). DIC was the most frequently observed C species (although much of the DIC database was indirectly derived from alkalinity data), with two thirds of DIC sites concentrated in western Australia (Figure 1A). The distribution of DOC was skewed towards eastern Australia (62%), where most concurrent measurements
of DIC and DOC were recorded. Sites with concurrent $CO_2$ and $CH_4$ observations mostly occurred in northern (45%, 10°S-20°S) and southern Australia (32%, 35°S-45°S), with 96% of these sites stemming from our field campaigns.

3.2.  Temporal data coverage

The temporal coverage of C data at each site varied considerably, ranging from a single observation at 1,368
sites (i.e. 47%) to a maximum of 1,307 observations at one site (Figure 2A and 2B). $CO_2$ was more commonly found at sites with a single observation, whereas $CH_4$ showed similar prevalence at sites with both single and multiple observations (Figure 2A and 2B). The longest DIC record included 1,111 observations over 20 years (ID: 129001A, WA) and the longest DOC record included 1,173 observations over 35 years (ID: 401201, VIC). The longest $CO_2$ record had 401 observations over 26 years (Murray River at Euston Weir, Bargrizan et al., 2022) and
the longest $CH_4$ record had 20 observations in one year (Rocky Mouth Creek, Gatland et al., 2014).

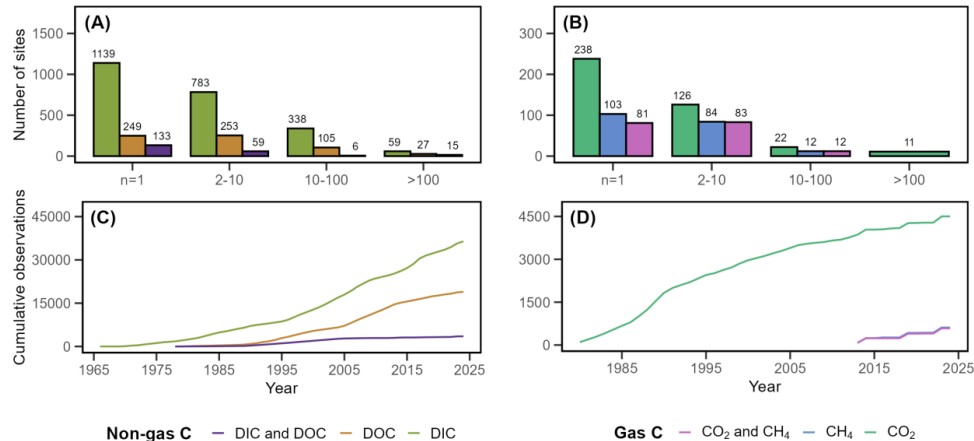

Figure 2. (Left) Non-gas C (DIC and DOC), and (right) gas C ($CO_2$ and $CH_4$) concentrations, grouped by (A, B) the cumulative number of annual observations from 1966 to 2024, and (C, D) the number of repeated observations per site.

The data in OzRiCa covered the period from 1966 to 2024 (Figure 2C and 2D). DIC observations spanned

the entire period, while DOC, $CO_2$ and $CH_4$ measurements began in 1978, 1980 and 2013, respectively. Concurrent measurements of non-gas (Figure 2C) or gas C species (Figure 2D) started after 1978 and 2013, respectively, with $CH_4$ mostly measured together with $CO_2$, while DOC was frequently measured independently of DIC.

Monthly observations revealed that sampling was more consistent and regular across all climatic regions for non-gas C sites (Figure 3A).

Mediterranean and semi-arid regions had the highest number of non-gas C observations, driven by extensive state agency monitoring in WA, which contributed 79% of Mediterranean observations, while 89% of semi-arid observations were sourced from the VIC and NSW state databases. In contrast, for gas C sites, consistent and regular sampling was limited to arid and semi-arid regions, with relatively frequent sampling in tropical and subtropical sites, and very irregular and less frequent sampling in temperate and Mediterranean sites (Figure 3B).

The strong temporal coverage of gas C in arid and semi-arid regions was in large part due to a long-term database from the Murray River that provided monthly observations over 26 years (Murray River at Cobram; Bargrizan et al., 2022) and 32 years (Murray River at Tailem Bend; Bargrizan et al., 2022). In contrast to other climatic regions which had relatively even numbers of $CO_2$ and $CH_4$ observations, at least 99% of observations in arid and semi-arid regions were $CO_2$ rather than $CH_4$ concentrations. The subtropical and tropical regions had observations in

nearly every month (Figure 3B), reflecting the high number of sites in these regions (Figure 1D; Table 1). However, the data were unevenly distributed, because most observations were from different studies. Peaks in gas C observations occurred in February, March, and September, coinciding with our first two sampling campaigns. In Mediterranean regions, where the number of sampled sites was high (Table 1), more than half of the months lacked gas C concentration data (Figure 3B). Although the site density was one of the highest for temperate regions (Table

1), the number and frequency of observations remained limited, with only nine sites having more than one gas C observation. November stood out as the month with the highest number of observations and the most concurrent observations, due to our last sampling campaign that captured both C species in temperate regions.

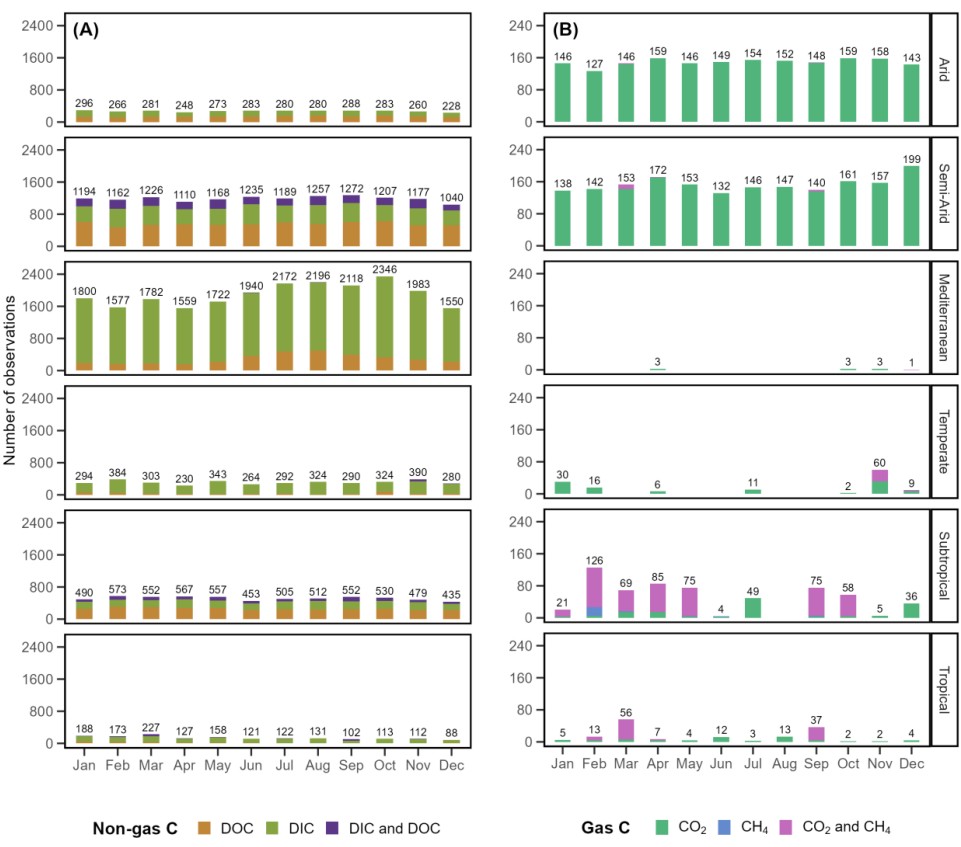

Figure 3. Number of observations of (A) non-gas C (DIC and DOC), and (B) gas C ($CO_2$ and $CH_4$), grouped by month and climate.

Although spot observations were the most common observation across all climatic regions, repeated sampling was frequent across sites (Figure 4), with nearly half of the sites having more than two measurements (Table 1). Among seasonal sites, only 21 sites had 12 or more observations. In contrast, 193 interannual sites had records extending over 10 or more years, with 94% sourced from state agencies in NSW, VIC, WA and VIC. Seasonal and interannual sites primarily covered Mediterranean and semi-arid regions, with the latter corresponding, for instance, to the long-term study of the Murray River (Bargrizan et al., 2022).

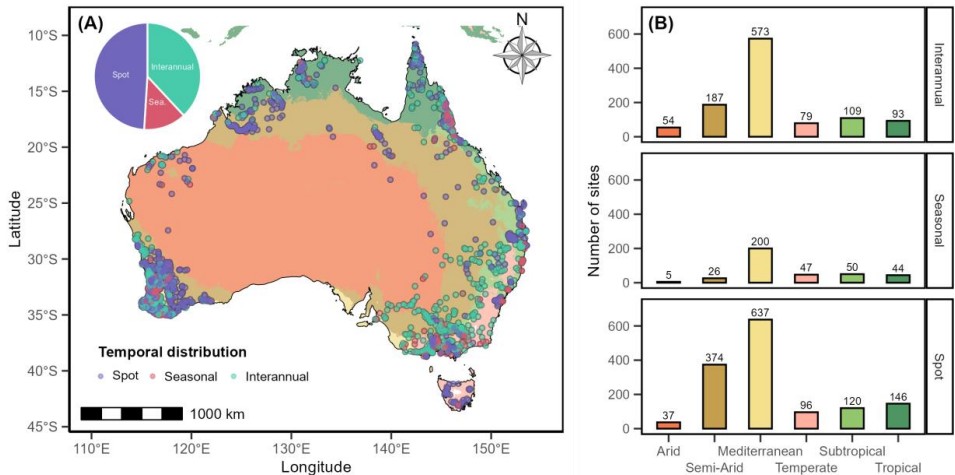

Figure 4. (A) Spatial distribution and (B) measurement frequency of sites in OzRiCa. The colours on the map and bar plot represent different climatic classes, while the dot colours on the map correspond to temporal categories.

### 3.3. Overview of concentration, flux, and isotope C data

The C data in OzRiCa, including different species, isotopes, and fluxes, spanned several orders of magnitude (Table 2, Figure 5). Among C concentrations, DIC had the highest median values, followed by DOC, $CO_2$ and $CH_4$. DIC and $CO_2$ observations obtained directly versus indirectly were significantly different (p=0.000105 for DIC and $p<10^{-15}$ for $CO_2$; Wilcoxon rank-sum test), with median direct $CO_2$ concentrations 125% higher than median indirect $CO_2$ concentrations, while median direct DIC concentrations were 9% higher than median indirect DIC concentrations (Table 2). The large difference for $CO_2$ estimates arises because 92% of indirect $CO_2$ data in the database are derived from large rivers (stream orders 7-9) (e.g. Bargrizan et al., 2022), whereas the direct $CO_2$ measurements include 78% of observations from small streams and rivers (stream orders 1-3), which typically have higher gas concentrations. Isotopic compositions showed a large range, with $CH_4$ showing the most depleted values (median -50.5‰) and DIC the most enriched (median -9.8‰). The isotopic values of $CO_2$ (median -21.1‰) was slightly higher than those of DOC (median -25.1‰). The median atmospheric flux of $CO_2$ was in molar terms 272 times higher than for $CH_4$ (Table 2). Among C species, $CH_4$ concentration and to a lesser extent $CO_2$ concentration, exhibited the highest coefficients of variation (CV), while DIC showed the greatest CV among stable C isotopes (Table 2). Most distributions were unimodal, but gas C fluxes and $CH_4$ concentrations (i.e. the species with fewer observations) had bimodal distributions (Figure 5).

Table 2. Summary of statistics for C concentrations, fluxes and isotopes for stream waters. Obs., Sites, SD, and CV refer to number of observations, number of sites, standard deviation, and coefficient of variation, respectively.

| Metric | C Type | Sites | Obs. | Mean | Median | Min | Max | SD | CV |
|---|---|---|---|---|---|---|---|---|---|
| Concentration ($\mu$mol L$^{-1}$) | DIC direct | 157 | 497 | 1428.7 | 1210.0 | 30.0 | 7170.0 | 1300.6 | 91.0 |
| | DIC indirect | 2169 | 35903 | 1170.1 | 1114.9 | 0.0 | 19801.0 | 1310.0 | 111.9 |
| | DOC | 634 | 18941 | 732.7 | 416.7 | 8.3 | 37500.0 | 990.2 | 135.2 |
| | $CO_2$ direct | 192 | 677 | 194.2 | 163.7 | 14.7 | 3322.0 | 239.7 | 123.4 |
| | $CO_2$ indirect | 210 | 3826 | 111.8 | 72.8 | 1.6 | 11382.1 | 270.6 | 242.1 |
| | $CH_4$ | 199 | 612 | 2.0 | 0.6 | 0.0 | 81.3 | 7.2 | 360.0 |
| Isotope (‰ vpdb) | DIC | 235 | 609 | -11.0 | -9.6 | -28.4 | 0.2 | 5.8 | 52.7 |
| | DOC | 233 | 407 | -25.5 | -25.2 | -43.8 | -20.0 | 3.1 | 12.2 |
| | $CO_2$ | 123 | 123 | -21.2 | -21.1 | -28.3 | -12.1 | 3.8 | 17.7 |
| | $CH_4$ | 173 | 223 | -50.3 | -50.5 | -113.7 | 14.6 | 10.4 | 20.7 |
| Flux (mmol m$^{-2}$ d$^{-1}$) | $CO_2$ | 160 | 478 | 411.6 | 244.8 | 1.8 | 7505.0 | 750.2 | 182.3 |
| | $CH_4$ | 129 | 520 | 3.0 | 0.9 | 0.0 | 73.3 | 7.0 | 233.1 |

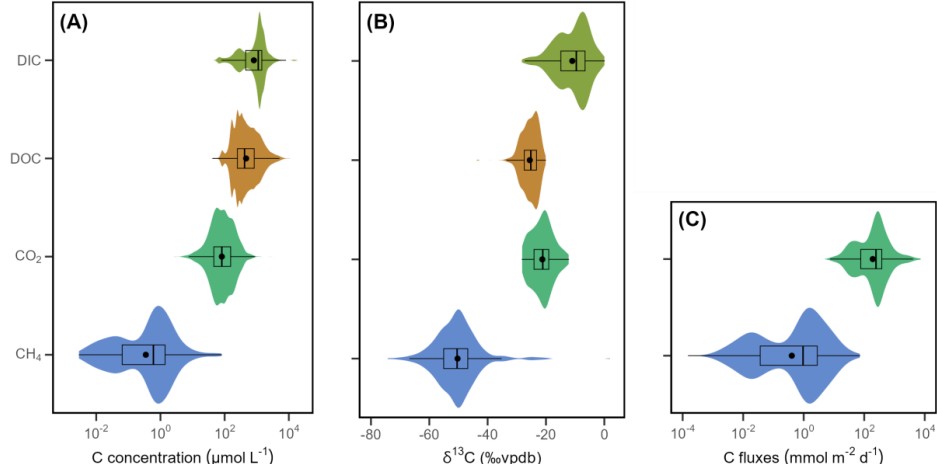

Figure 5. Box and violin plots of the gas and non-gas C concentrations (A), stable isotopes (B), and fluxes (C). Violin plot width is scaled to the number of observations of each individual C species. Black dot and vertical lines within the boxplot represent the mean and median values. The upper and lower edges of each box are the 25$^{th}$ (Q1) and 75$^{th}$ (Q3) percentiles. The minimum/maximum whisker values are calculated as Q3/Q1 ± 1.5*(Q3-Q1). Any values beyond the whiskers are considered outliers.

No consistent increase or decrease in the magnitude of C species across stream orders was observed, likely due to the high variability between sites within the same stream order (Figure 6). Low-order streams (1-3) had significantly lower median DIC concentrations than high-order streams (7-9) (p<0.01, Wilcoxon rank-sum test). A similar trend was observed for $CH_4$, which showed significantly higher median values with increasing stream order (1-5, p<0.01, Kruskal-Wallis test); however, the absence of $CH_4$ data from high-order streams limited this analysis. DOC and $CO_2$ concentrations had two distinct peaks, one for low to medium-order streams (2-4) and the



other one for high-order streams (7-9), with 99.7% of $CO_2$ concentrations in the latter derived through indirect estimation methods (mostly from various sites along the Murray River). Overall, median values of DIC and DOC isotopes, along with C gas fluxes, appeared to increase with stream order (1-9 for dissolved and 1-5 for gas C, $p < 0.01$, Kruskal-Wallis test).

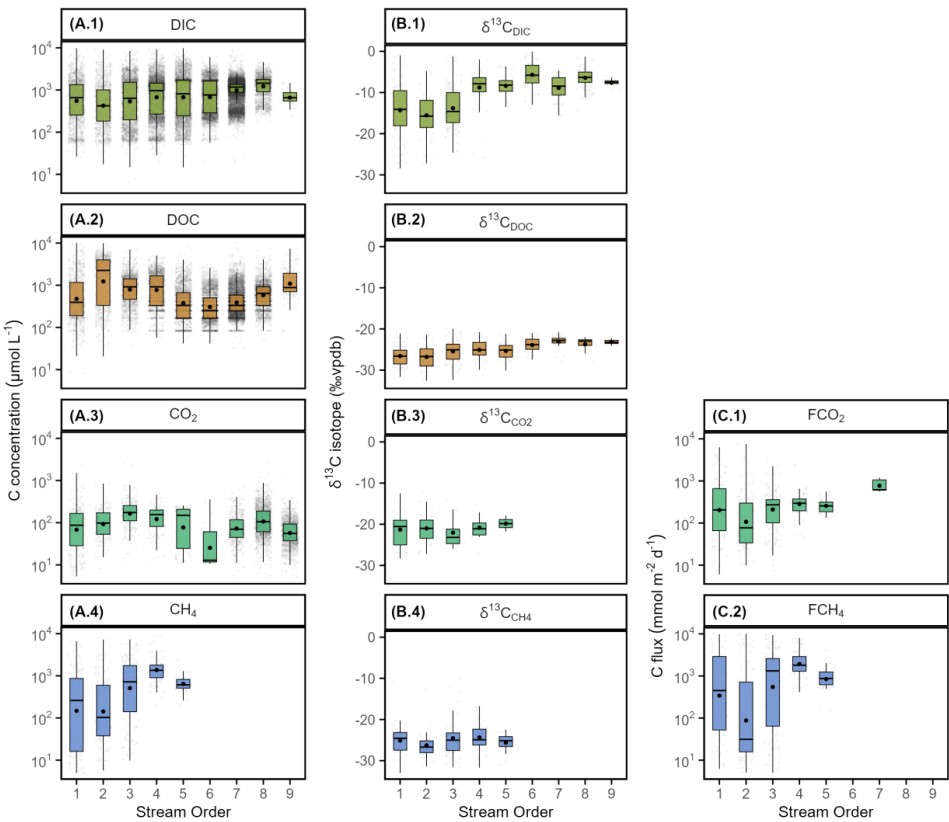

Figure 6. (A) C concentrations, (B) isotopic composition, and (C) gas fluxes as a function of Strahler stream order. The units for $CH_4$ differ from those used for other C species, shown in nmol $L^{-1}$ in (A.4), ‰vpdb/2 in (B.4), and nmol $m^{-2}$ $d^{-1}$ in (C.2). Black dot and vertical lines within the boxplot represent the mean and median values. The upper and lower edges of each box are the 25th (Q1) and 75th (Q3) percentiles. The minimum/maximum whisker values are calculated as Q3/Q1 ± 1.5*(Q3-Q1). Any values beyond the whiskers are considered outliers.



### 4. Discussion

OzRiCa gathers data from a wide range of sources including 64 publications, two theses, five government agencies, and new data being published for the first time into a single, cohesive dataset. Somewhat unique about OzRiCa is its broad spatial coverage across diverse climates and stream orders, and its focus on all major riverine C species. Building OzRiCa required a meticulous compilation and synthesis of observations across a geographically, climatically and ecologically diverse continent. We hope this database can serve as a critical resource for advancing C cycling in Australia and globally. Recently, aggregated datasets similar to this one have been instrumental in advancing riverine C cycling research, including large scale flux modelling (e.g. Rocher-Ros et al., 2023; Stanley et al., 2023), cross-site comparisons (e.g. Bernhardt et al., 2022; Gómez-Gener et al., 2021), and the establishment of scaling relationships for gas exchange research (e.g. Aho et al., 2024; Raymond et al., 2012). Beyond the C research community, we believe OzRiCa has potential applications in other disciplines, as riverine C plays a key role in ecosystem functioning and nutrient cycling within the critical zone.

While OzRiCa represents a significant advancement, this database also makes clear the current gaps in national-scale riverine C data, which must be addressed to better understand the spatial and temporal variability of different C species in streams and rivers across Australia. By highlighting these gaps, the database provides an opportunity to identify where, when and how future research and monitoring should be directed. Below, we examine the spatial and temporal distribution of sampling efforts, discuss relevant methodological considerations, and conclude with potential opportunities offered by OzRiCa.

### 4.1. Improved spatial coverage in C observations

The growth of Australian riverine C studies over the past decade (Figure 2) has improved the nationwide data coverage, yet the distribution of sites remains uneven across the country (Figure 1). The recent geographic expansion mainly involves DIC observations in eastern and western Australia, largely driven by indirect records from different state agencies, along with our previously unpublished observations of non-gas and gas C in northern and southern Australia. Western Australia also has DOC observations from 151 sites recorded between 1991 and 2019, but gas C remains largely unmeasured in this region. OzRiCa further highlights the uneven distribution of observations, with areas far away from the coasts particularly underrepresented. A key challenge in these mainly remote areas is their predominantly arid and semi-arid environments (Table 1, Figure 4) that contribute to the prevalence of sparse and intermittent streams and rivers. These systems play a significant role in C and nutrient cycling, ecosystem function, food security (Shanafield et al., 2024) and C budgets (Stanley et al., 2023). For instance, recent studies have shown that including seasonal drying and rewetting of riverbed sediments increase global inland water C emissions (e.g. López-Rojo et al., 2024; Marcé et al., 2019; Qin et al., 2024). Adding data from arid and semi-arid regions will help improve our understanding of C cycling in intermittent systems, which drain over 70% of Australia's river network (Sheldon et al., 2010) and have become more prevalent over recent decades (Sauquet et al., 2021). While tropical regions are also underrepresented in OzRiCa, this study includes previously unpublished data that improves their geographic representation. Tropical inland waters are recognised as significant hotspots for gas C emissions, and the lack of data in this region remains a global issue (Drake et al., 2018; Lauerwald et al., 2023). These climatic regions, which are still poorly studied, face growing pressure from rising human land and water demands (e.g. Duvert et al., 2022; Stringer et al., 2021) and climate change (e.g. Lian et al., 2021), underscoring both the need and the opportunity for targeted future monitoring.



The aggregation of data from numerous studies into OzRiCa has provided coverage across most stream orders, with low-order streams (1-3) being the most prevalent in the database (Figure 1). This focus on low-order streams is particularly important, given their abundance across river networks (Downing et al., 2012; Marx et al., 2017), their sensitivity to anthropogenic and environmental changes (Ulloa-Cedamanos et al., 2024), their underrepresentation in global databases (Drake et al., 2018; Liu et al., 2024), and their relevance for total riverine

C emissions, despite covering only a small percentage of the catchment surface area (e.g. Ågren et al., 2007; Marx et al., 2017; Natchimuthu et al., 2017; Wallin et al., 2018). High-order streams (7-9) play a crucial role in the downstream export of non-gas C species to the ocean. Although OzRiCa includes DIC and DOC observations from major rivers (Figure 6), our analysis underscores the need for additional $CH_4$ concentration observations, as well as stable C isotopes and atmospheric C emission in large river systems, where significant gaps still exist.

4.2.   Addressing temporal variability

     Unlike the spatial coverage of Australian riverine C data, which has improved over the last decade, most sites still suffer from limited temporal coverage. The predominant sampling strategy involves collecting one or a few samples from individual sites, resulting in short and sparse time series (Liu et al., 2024). This limited temporal coverage is specifically reflected in OzRiCa for C gas records, where only 32 sites had more than 10 observations,

with temperate regions particularly low in time-series data, as most sites had just one or two observations (Figure 3, Table 1). For non-gas C records, the situation is better with 476 out of 2,744 sites having more than 10 observations (Figure 2), largely due to monitoring by state agencies. Long-term data (>5 years) are extremely infrequent for gas C and infrequent for non-gas C. These data come from a large-scale study in the arid and semi-arid region of the Murray-Darling Basin ((Bargrizan et al., 2022) and from state agencies. The Murray-Darling

Basin study focused largely on DOC and $CO_2$ observations in the Murray River, providing river-specific insights that might not be applicable to smaller streams and rivers of the arid and semi-arid regions. No such long-term data exist for $CH_4$, with the longest time series being up to two years (Atkins et al., 2017). Despite increasing efforts to address these temporal gaps, most studies are limited to a few years, as very few funding programs support long-term monitoring. The limited long-term observations are further challenged by the current downward

trend in the number of long-term discharge monitoring stations (Shanafield et al., 2024), which are crucial for estimating C fluxes and understanding C dynamics.

     The lack of data at finer temporal scales can lead to inaccurate C flux estimations, as critical C processes may be overlooked. For example, predominant sampling during daytime, excluding diel variability, may result in a consistent underestimation of global riverine GHG emissions, since nighttime $CO_2$ emissions are on average 30%

higher than daytime emissions (Gómez-Gener et al., 2021). Integrating hydrological events is pivotal to better constrain C export (Lauerwald et al., 2023), as more than half of the annual DOC export can occur during the early flood stages (e.g. Birkel et al., 2020; Ulloa-Cedamanos et al., 2021a). Australia's extreme hydroclimate variability, which has intensified over the past decades (Ayat et al., 2022), fosters these hot moments of C export. The previously identified data limitations underscore the need for high-resolution and long-term data to capture

these critical temporal variations and improve the accuracy of C flux estimates across different climatic regions.

4.3. More challenges and opportunities

As discussed above, the increase in the availability of both gas and non-gas riverine C data has been remarkable, creating new opportunities to examine large-scale spatial and temporal patterns. However, specific challenges in C observations become evident when aggregating different studies together in a single database. For instance, gas flux estimates require the inclusion of different emission pathways, particularly for the $CH_4$ flux. Currently, OzRiCa only includes observations from the diffusive $CH_4$ pathway, although the ebullitive $CH_4$ flux can be substantial in riverine systems (Stanley et al., 2023). The analysis of C species also involves multiple steps and calculations, with field and laboratory protocols varying widely within the same C species. Although direct measurements of DIC and $CO_2$ should be prioritised, indirect estimations are still observed in environmental agency databases (Hartmann et al., 2014) and from the literature, particularly in studies using long-term data (e.g. Binet et al., 2020; Rosentreter and Eyre, 2020; Stets et al., 2014; Ulloa-Cedamanos et al., 2020). OzRiCa is not an exception, with indirect DIC data representing a large part of the DIC data, nearly all of which was sourced from state agencies. While indirect DIC calculations are generally reliable within the typical pH range of rivers, indirect $CO_2$ estimates can introduce significant errors in systems with low buffering capacity and high DOC concentrations (Abril et al., 2015; Liu et al., 2020). In OzRiCa, nearly all the indirect $CO_2$ concentrations come from eight peer-reviewed publications (with 90% of these data from Bargrizan et al., (2022)), where we assumed the authors had ensured the accuracy and reliability of their indirect estimates. A few other indirect $CO_2$ estimates come from our sampling campaigns and were included only when direct measurements were unavailable or deemed unreliable (see further details in Appendix A.2). Although we observed a statistically significant difference between the median direct and indirect DIC concentrations, the magnitude of this difference was relatively small (9%). The median direct and indirect $CO_2$ observations differed more substantially (125%), but this was related to a bias in stream order between the two groups, with small streams – typically exhibiting higher $CO_2$ concentrations (Marx et al., 2017; Wallin et al., 2018) – overrepresented in the direct dataset. Based on these considerations, we have high confidence in the robustness of both direct and indirect $CO_2$ and DIC estimates included in OzRiCa.

Significant gaps in C databases include the limited number of isotopic C data and the lack of concurrent measurements of multiple C species (DIC, DOC, POC, $CO_2$, and $CH_4$) at the same sites. Isotopic data are key for identifying biogenic and geogenic sources of C (e.g. Campeau et al., 2017), in-stream processes and residence times of C. For instance, $\delta^{13}C$ isotopes can help pinpoint $CO_2$ sources (Duvert et al., 2020b; Telmer and Veizer, 1999), which are often assumed to originate primarily from respiration in wetlands and upland soils (Liu et al., 2022c). Concurrent measurements of multiple C species are also necessary to better understand the mechanisms driving temporal and spatial variations in C sources and sinks. While our three sampling campaigns address this issue in OzRiCa, most other studies in the literature focus on either DOC or $CO_2$, with DIC and $CH_4$ often observed sporadically. In contrast, state agencies typically focus on either DOC or indirect DIC records. Including isotopic C and multiple species of C in future studies could offer deeper insights into the sources and processes of C in large-scale assessments.

OzRiCa represents a powerful tool for exploring the role of rivers in the global C cycle, functional ecology and ecosystem health of inland waters. This database provides a better understanding of spatial and temporal dynamics of C species in Australian rivers and streams, with particular strengths in addressing areas that were historically underrepresented, such as tropical and semi-arid areas, and low-order streams. These areas, critical in



C cycling due to their sensitivity to anthropogenic and environmental changes, have been prioritised in this effort. We hope OzRiCa will help improve our comprehension of freshwater C processes and better prepare us to predict their responses to global environmental changes. OzRiCa also offers unique opportunities through the integrated analysis of C data (Table A) and basin-specific data (Table B), enabling the parameterisation of freshwater C

models and the development of scalable products for regional and continental assessments. This approach allows researchers to benchmark model outputs against real observations from Australia, providing a robust framework for advancing our understanding of C dynamics in freshwater systems.

**Data availability**

The OzRiCa database is shared freely on Hydroshare, and data are available for download at

http://www.hydroshare.org/resource/9aa735254e7e424ca18603c047d02f50 (Ulloa-Cedamanos et al., 2025). The database is stored in comma-separated values (CSV) file format and include the following tables: Table A. OzRiCa carbon data, Table B. OzRiCa basin-specific data, and Table C. OzRiCa references. R scripts that were used to pre-process data into inputs are available upon request to the authors. The dataset is released under the Creative Commons Attribution 4.0 International license.

**Author contributions**

F.U.-C.: Writing- original draft, Conceptualization, Methodology, Data Collection, Software development, Data curation, Formal validation analysis, Visualization, Writing- Review and Editing; A.T.R.: Writing- original draft, Conceptualization, Methodology, Data Collection, Software development, Data curation, Writing- Review and Editing; Y.L.: Conceptualization, Data Collection, Writing- Review and Editing; L.B.H.: Data Collection,

Writing- Review and Editing; W.W.W.: Data Collection, Writing- Review and Editing; M.W.: Writing- Review and Editing; J.G.C.: Writing- Review and Editing; A.L.: Writing- Review and Editing; C.D.: Writing- original draft, Conceptualization, Methodology, Data Collection, Data curation, Writing- Review and Editing, Funding acquisition, Supervision, Resources.

**Competing interests**

The authors declare that they have no conflict of interest.

**Acknowledgements**

Our thanks go to Jackie Webb, Ryan Burrows, Luke Mosley, Paul Nelson, Aleicia Holland, Tim Wardlaw, Ewen Silvester for providing data and theses. We also acknowledge the following Australian state departments, source and custodian of their data, for granting us permission to include their data in this database: New South

Wales Government, WaterNSW; Northern Territory Government, Department of Lands, Planning and Environment; Queensland Government, Department of Environment, Science and Innovation; Victoria Government, Department of Energy, Environment and Climate Action; Western Australia Government, Department of Water and Environmental Regulation. We extend special thanks to Niels Munksgaard, Matt

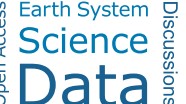

Northwood, Adam Bourke, Hao Wang, Michael Liddell, and Sophie Golding-Chan for their contributions before, during and after the fieldwork campaigns and laboratory analyses, and to David Butman for discussions. This research has been supported by the Australian Research Council (DP220100823, DE220100852).



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



## Appendix


Appendix A. Detailed field methods

A.1 Data collection

Length and channel slope were determined through a simple topography survey using an automatic levelling laser (NA2 Automatic level; Leica Geosystems). At each site, channel width was measured at 2–4 transects spaced

along the study reach, perpendicular to the length of the stream. At each transect, depth was measured every 10–50 cm. The reported widths are the average of all transects and the reported depths are the average of all measurements across all transects. Water temperature, dissolved oxygen, specific conductivity, and pH were measured using either a YSI ProSolo equipped with temperature and conductivity probes, or a YSI EXO2 multiparameter sonde equipped with an optical DO sensor, conductivity and temperature sensor, and pH sensor.

Flow rate and velocity were measured using either salt dilution gauging or a flow meter (OTT MF Pro - Water Flow Meter; OTT HydroMet). For salt dilution gauging, a salt slug of 50–1000g (depending on stream size) was introduced upstream of the sampling reach, ensuring full mixing between the injection and the measurement points. Specific conductivity was measured at both the upstream and downstream ends of the reach using the YSI ProSolo and YSI EXO2. Discharge was calculated by dividing the mass of salt by the area under the baseline-

corrected specific conductivity timeseries converted to salt mass concentration using a conversion factor of 0.5 mg L$^{-1}$ per μS cm$^{-1}$ (Richardson et al., 2017). Velocity was calculated by dividing the distance between the two sensors by the time it took for the salt slug to travel from the upstream to the downstream sensor. This time was estimated as the time difference between the two conductivity peaks.

The flux measurements from the floating chambers were also used to calculate the gas transfer velocity of

$CO_2$ ($k_{CO2}$). Average $CO_2$ flux of the site was divided by the difference between the $CO_2$ concentration in the stream and in the atmosphere. For consistency, this gas transfer velocity was converted to a standard gas transfer velocity for $CO_2$ at 20°C ($k_{600}$) using the Schmidt number (Eq. A1 where $T$ is temperature °C) and Eq. A2.

$$Sc_{(CO_2)} = 1911 - 118.11T + 3.453T^2 - 0.0413T^3 \tag{A1}$$

$$k_{600} = \left(\frac{600}{Sc_{(CO_2)}}\right)^{-0.5} \times k_{CO_2} \tag{A2}$$

In some cases when the stream was too turbulent to use a floating chamber, gas transfer velocity was measured using a propane tracer gas. In these instances, propane was diffused into the stream using an

approximately 10cm diameter aquarium stone placed upstream of the study reach. After 30-40 minutes, eight water samples (four upstream and four downstream) were collected in 20 mL glass vials sealed with aluminium crimp caps and septas. The samples were collected in transects across the stream and without any gas headspace. Samples were refrigerated until analysis at Charles Darwin University where, just prior to analysis, 10 mL of water was replaced with atmospheric air (assumed to have no or negligible propane) to create a 1:1 water:

headspace mixing ratio. Samples were analysed on a Clarus 680 gas chromatograph (Perkin Elmer) with a flame ionization detector (FID). Gas transfer velocity was calculated using Eq. A3.

$$k_{(C_3H_8)} = \frac{-log\left(\frac{C_x}{C_0}\right) vz}{x},$$
(A3)

where $C_0$ and $C_x$ are the concentration of propane at the up and downstream, respectively, $x$ is the distance of the stream reach (m), $v$ is the flow velocity (m s$^{-1}$), and $z$ is the depth (m). This propane-specific $k$ value was converted to $k_{600}$ using the Schmidt number for propane (Eq. A2 and Eq. A4), substituting $Sc_{(CO2)}$ and $k_{CO2}$ with $Sc_{(C3H8)}$ and $k_{(C3H8)}$.

$$Sc_{(C_3H_8)} = 2864 - 154.14T + 3.791T^2 - 0.0379T^3$$
(A4)

A.2 Data selection

For each site, we report a single value for each parameter, even though multiple methods were used to measure several variables at the same site. The reported value was selected based on the method with the highest confidence. Methods were ranked in order of confidence, and the value from the highest-ranked method available for each site was used.

$CO_2$ and $CH_4$ concentrations measured using the headspace technique ('method 1') were given the highest priority, followed by concentrations obtained from $HgCl_2$-preserved samples ('method 2'). The third method, which involved the indirect calculation of $CO_2$ from DIC, temperature and pH ('method 3'), was ranked lowest and employed only when the other two methods were unavailable. Method 2 was preferred over method 1 if the coefficient of variation between replicate headspace measurements was higher than 20%. Method 3 was preferred over method 2 if the pH was above 6 and the $CO_2$ concentration from method 2 was higher than that from method 3. This decision workflow reflects the evidence suggesting that $HgCl_2$ preservation may artificially increase measured $CO_2$ and $CH_4$ values, particularly in higher pH systems (>7) (Clayer et al., 2024). Overall, 64% of our $CO_2$ data were obtained from method 1, 7% from method 2 and 29% from method 3. For DIC concentrations, direct DIC measurements from $HgCl_2$-preserved samples were ranked highest, followed by values calculated indirectly using $CO_2$, temperature, and pH.

Flow velocity was reported using the salt gauging velocity value if available, as this method provides a measurement that reflects the entire stream reach, unlike a flow meter, which captures velocity at a single point. We prioritised flow rate values obtained from salt gauging using the EC sensor that produced the smoothest breakthrough curve, followed by data from the second EC sensor, followed by flow meter measurements if salt gauging data were unavailable. In most cases, the best breakthrough curve came from the downstream sensor.



Appendix B. OzRiCa metadata.

Table B.1. Column titles and description of their content for the OzRiCa C data table.

| Parameter | Description | Unit |
|---|---|---|
| Ref | Reference # to paper/thesis/state databases as per '3.DataReferences.csv' | Alphanumeric |
| ID | Name of site. Either the name of the stream, road crossing, or internal state code. | Alphanumeric |
| Date | Date of sampling. | YYYY-MM-DD |
| Time | Time sampling began. | HH:MM:SS |
| Latitude | Latitude measurement taken in the field. Geographic coordinate system: WGS84. | Decimal Degrees |
| Longitude | Longitude measurement take in the field. Geographic coordinate system: WGS84. | Decimal Degrees |
| DIC | Dissolved inorganic carbon concentration. | $\mu mol\ L^{-1}$ |
| DIC.Method | Method to measure DIC, either direct or indirect. | Alphanumeric |
| DOC | Dissolved organic carbon concentration. | $\mu mol\ L^{-1}$ |
| $CO_2$ | Dissolved carbon dioxide concentration. | $\mu mol\ L^{-1}$ |
| $CO_2$.Method | Method to measure $CO_2$, either direct or indirect. | Alphanumeric |
| $CH_4$ | Dissolved methane concentration. | $\mu mol\ L^{-1}$ |
| POC | Particulate organic carbon concentration. | $\mu mol\ L^{-1}$ |
| DIC.C13 | Delta thirteen ($\delta^{13}C$) of DIC. | ‰vpdb |
| DOC.C13 | Delta thirteen ($\delta^{13}C$) of DOC. | ‰vpdb |
| CO2.C13 | Delta thirteen ($\delta^{13}C$) of $CO_2$. | ‰vpdb |
| CH4.C13 | Delta thirteen ($\delta^{13}C$) of $CH_4$. | ‰vpdb |
| $FCO_2$ | Dissolved carbon dioxide emission flux. | $mmol\ m^{-2}\ d^{-1}$ |
| $FCH_4$ | Dissolved methane emission flux. | $mmol\ m^{-2}\ d^{-1}$ |

Table B.2. Column titles and description of their content for the OzRiCa basin-specific data table.

| Parameter | Description | Unit |
|---|---|---|
| ref | Reference # to paper/thesis/state databases as per '3.DataReferences.csv' | Alphanumeric |
| ID | Name of site. Either the name of the stream, road crossing, or internal state code. | Alphanumeric |
| Date | Date of sampling. | YYYY-MM-DD |
| Time | Time sampling began. | HH:MM:SS |
| Latitude | Latitude measurement taken in the field. Geographic coordinate system: WGS84. | Decimal Degrees |
| Longitude | Longitude measurement take in the field. Geographic coordinate system: WGS84. | Decimal Degrees |
| Strahler | Reported Strahler stream order at the reach where the sample was taken. | |
| Climate.Name | Name of climatic region at the reach where the sample was taken. | Alphanumeric |



| Catchment.Area | Reported basin size. | Km$^2$ |
|---|---|---|
| Temperature | Reported water temperature. | Degree Celsius |
| SpConductivity | Reported specific conductivity. | µS cm$^{-1}$ |
| pH | Reported pH. | |
| DO.MGL | Reported dissolved oxygen concentration. | mg L$^{-1}$ |
| DO.pct | Reported percent saturation of dissolved oxygen. | % |
| Length | Reported stream channel length. | m |
| Depth | Reported stream channel depth. | m |
| Width | Reported stream channel width. | m |
| Slope | Reported stream channel slope. | m m$^{-1}$ |
| Q | Reported flow discharge. | L s$^{-1}$ |
| Q.Method | Method to measure flow discharge, either salt gauging, flowmeter or undefined. | Alphanumeric |
| V | Reported flow velocity. | m s$^{-1}$ |
| V.Method | Method to measure flow velocity, either salt gauging, flowmeter or undefined. | Alphanumeric |
| k600 | Gas exchange coefficient. | m d$^{-1}$ |
| k600.sd | Standard deviation of gas exchange coefficient. | m d$^{-1}$ |
| k600.method | Method to measure k600, either chamber or tracer gas. | Alphanumeric |

Table B.3. Column titles and description of their content for the OzRiCa reference table.

| Parameter | Description | Unit |
|---|---|---|
| Ref | Reference # to paper/thesis/state databases. | Alphanumeric |
| Authors | Authors last name followed by first name initials. | Alphanumeric |
| Title | Title of data source. | Alphanumeric |
| Journal | Identity of the outlet for the data (e.g. journal, or agency that presented the data) | Alphanumeric |
| Year | Year of publication or data acquisition of a public/unpublished database. | YYYY |
| DOI | DOI or hyperlink for journal article or other publication based on the data. | Alphanumeric |



Appendix C. Supplementary information.

Figure C1. Environmental relationships with site-average CO₂ (A-D), CH₄ (E-H), DIC (I-L) and DOC (M-P) concentration

versus concurrent measures of dissolved 40oxygen (O₂), specific conductivity (SpCond), water temperature (T), and pH.

Linear regressions indicate that dissolved oxygen and specific conductivity partly account for the variation in DIC and CO₂

among sites.