# Peer review of "OzRiCa: An Australian riverine carbon database of concentrations, gas fluxes and isotopes"

_Earth System Science Data, 2025_

## Author Comment (AC1)

**Reviewer 1**

This well-written manuscript presents a large, compiled dataset of carbon concentrations, fluxes, and isotopes for rivers in Australia. The dataset comprises DOC, DIC, POC, CO2, and CH4, while the manuscript describes DOC, DIC, CO2, and CH4. This dataset has the strong potential to support an analysis of riverine carbon fluxes for the Australian continent and includes new, unpublished data. Overall, I find the manuscript and dataset comprehensive and well described, and the dataset will likely support an important analysis of continental-scale carbon fluxes from a relatively understudied region. However, I have a few comments that should be addressed.

We acknowledge reviewer #1 for their assessment of our work. Our responses to their comments are in green below.

First, it might be worth re-considering not describing the POC data in the manuscript. I understand that the authors choose not to describe the POC data because of the relatively small number of samples compared to the other parameters (text says n=51, I'm seeing n=77 in the dataset). However, POC is a part of the dataset, exhibits a large range of values (4 orders of magnitude), and is a relatively unconstrained riverine carbon flux, so the data is valuable. If the authors don't want to make direct comparisons between POC and DOC/DIC/CO2/CH4 given the very different number of samples, they could include a description of the POC data in a separate sub-section. That being said, it might also be interesting to see POC in Table 2 and Figure 5.

We totally agree with this comment. We initially chose not to describe POC in detail due to the low number of samples and sites. Following the reviewer's suggestion, we will now include POC in Table 2, and add a new Figure C2 in Appendix C showing POC spatial and temporal coverage (analogous to Figures 1 and 2) and POC violin and box plots (similar to Figure 5). We will also include a short description of the POC dataset at the end of Sections 3.1 (referencing Figure C2) and 3.3 (referencing Table 2 and Figure C2). The total number of POC samples increased from 51 to 77 after including a new reference (Tanner and Eyre, 2020), and the text will be corrected accordingly.

Second, the prevalence of river types could be considered in the discussion of spatial coverage and representativeness of the carbon dataset. Figure 1, Table 1, and the associated text very clearly point out that arid rivers aren't well represented in the dataset. This point is important in understanding the dataset. However, I also wonder what proportion of streams and rivers in Australia are found in the arid interior? How different is the proportion of samples from arid rivers (~5%) versus from non-arid rivers compared to the proportion of arid to non-arid rivers in Australia?

Thank you for this valuable suggestion. We will add the total area of the Australian river network with their coefficient of variation across climatic regions in Table 1. These values will also be described in Section 3.1 and discussed in Section 4.1, allowing us to explicitly compare the proportion of samples from arid vs. non-arid rivers represented in OzRiCa to their actual river distribution across Australia.

**Other comments:**

Ln 134-135: This sentence should be revised. As far as I understand, the authors did not calculate CO2 from literature values of pH and alkalinity, but rather only used literature values of calculated CO2.

Correct. We did not calculate CO2 using pH and alkalinity; we only included indirect CO2 values when they were already calculated and presented in the original sources. We will rephrase the sentence accordingly.

Ln 182-183: Headspace equilibration samples for CO2 should be corrected for carbonate equilibria (Koschorreck et al., 2021). At the pH range of the dataset (1$^{st}$-3$^{rd}$ Quartile: 7-7.8), bicarbonate would be the dominant form of inorganic carbon, and not CO2, so this correction might be important. The authors have DIC and pH so they could apply this correction to the pCO2 values or show that it doesn't have much of an effect.

Good point – for samples with available DIC and pH, we will applied carbonate equilibrium corrections to headspace-derived CO2 measurements.

Ln 246: Needs to be reworded to make clear that this comparison refers to observations, not concentrations. E.g., "DIC and DOC measurements were the most prevalent measurements in the dataset, across all climatic regions"

We have revised the sentence as suggested.

Ln 268-9: combine this stand-alone sentence into the next paragraph.

We have made the suggested change.

Figure 2 – It's difficult to see the colors in the legend – consider making the lines in the legend thicker.

We will increase the line thickness in the legend.

Figure 4a – It's difficult to read the words in the pie chart.

We will increase the font size in the pie chart labels.

Ln 306 – "The large difference for CO2 estimates *may arise* because…"

We have rephrased the sentence accordingly.

Ln 848-861 – How were the study-reach lengths choosen? What were the range of reach lengths? Were there any groundwater inputs or lateral flows along the study reaches? If so, were dilution corrections applied?

For the 144 sites where we measured either velocity or gas transfer velocity (k), study-reach lengths ranged from 6 to 84 m, with a mean of 21 m. Reach lengths were selected to capture the main hydraulic features (i.e., pools and riffles) and to ensure reliable salt dilution gauging, i.e. sufficient upstream mixing and a single, non-braided channel. For the twelve sites where we used propane tracer gas to estimate k, reach lengths ranged from 11 to 40 m (median = 18 m). No groundwater or surface inflows were observed along these reaches. Because sites with any visible inflows were avoided; we did not apply any dilution corrections for tracer gas or salt dilution measurements.

Ln 866-876 – Generally direct measurements of CO2 are much better than calculations, given all the errors associated with pH measurements and non-carbonate alkalinity. Why were the headspace replicates so frequently (36% of samples) off by > 20%?

Thank you for pointing this out. Across the entire dataset, coefficient of variation (CV) greater than 20% occurred in only 7% of samples (n = 13). The 36% value you refer to assumes that any CO2

measurement not classified as a headspace measurement (Method 1) must have exceeded a 20% of CV. However, Method 1, 2, and 3 were not always applied at every site. For example, during one of our sampling campaigns (73 measurements), $CO_2$ and $CH_4$ concentrations were determined exclusively using $HgCl_2$-preserved samples (Method 2), rather than by headspace technique (Method 1). Therefore, the proportion of headspace replicates exceeding 20% CV is much lower than inferred. We have clarified this point in the revised text.

Ln 877-881 – Aren't estimates of velocity derived from a pair of breakthrough curves/EC sensors? How can only one break through curve be used?

Sorry for the confusion. Lines 877-878 refer to flow velocity, while lines 879-881 refer to flow rate (discharge). Velocity was determined using a pair of EC sensors, and discharge was estimated from a single breakthrough curve. We have revised the paragraph for clarity.

Figure C1 – difficult to read the text on this figure.

We will increase the font size in the figure.

Ln 890 – extra "40" before oxygen.

Thanks, we have deleted this extra word.

Dataset: There are two extremely high [O2] values in the dataset (37.4 and 98.6 mg/l)

We agree with your observation. We have corrected one value (based on % DO saturation, the 37.4 mg/L value was likely a typo in the original paper and we replace it with 3.74 mg/L) and removed the other one.

Dataset: More than 6,000 entries don't have a catchment area. It should be possible to delineate a catchment using a DEM and estimate catchment area.

In fact, fewer than 2,000 sites in our dataset lack a catchment area. These correspond exclusively to sites compiled from the literature. Catchment areas were delineated for all sites sampled directly by us (fewer than 200). From that experience, we found that deriving reliable catchment areas requires extensive QA/QC beyond simple DEM-based delineation, especially for small and low-relief streams. We chose not to include automatically derived catchment areas to avoid introducing potentially large spatial errors. We prefer to maintain data quality and transparency rather than add uncertain estimates.

**References:**

Koschorreck, M., Prairie, Y. T., Kim, J., & Marcé, R. (2021). Technical note: CO2 is not like CH4 – limits of and corrections to the headspace method to analyse pCO2 in fresh water. *Biogeosciences*, *18*(5), 1619–1627. https://doi.org/10.5194/bg-18-1619-2021

**Author's references:**

Tanner, E. L. and Eyre, B. D.: Carbon Budget for a Large Drowned River Valley Estuary Adjacent to an Emerging Megacity (Sydney Harbour), Journal of Geophysical Research-Biogeosciences, 10.1029/2019JG005192, 2020.

---

## Author Comment (AC2)

**Reviewer 2**

I commend the authors on compiling and presenting this highly valuable and timely dataset of aquatic carbon parameters across Australia. The study addresses a significant gap in the global carbon cycle research, particularly for the under-sampled Australian continent. The collection of a harmonized dataset of Australian rivers, is a substantial achievement and represents a commendable effort. The manuscript is well-structured, the methodology for data collection and quality control appears robust, and the potential utility of this dataset is immense. It will undoubtedly serve as a critical benchmark and foundational resource for the modeling community, ecologists, and climate scientists. The authors have done an excellent job in making this dataset accessible, interoperable, and reusable) and the provision of detailed metadata and a data descriptor paper is highly appreciated. I therefore recommend a minor revision before it can be accepted for publication in ESSD.

We thank reviewer #2 for their evaluation of this work. Our responses to their comments are in green below.

**Line-to-line comments:**

Figure 1 and 4: since this manuscript mainly talk about riverine carbon, can you put major rivers in the maps.

We will add the major river networks to Figure 1 and Figure C1 for geographic and climatic context. However, we prefer not to include them in Figure 4 to not overload the figure with too much information.

Also can you pub climate zone here or in the supplementary information.

Climatic zones are already represented in Figure 4. Nevertheless, we are happy to add a new Figure C3 in Appendix C showing the climatic zones across Australia.

You also need to plot figures for POC observations as well. Can you put the information about POC in the SI?

Agreed. We will include POC in Table 2, and add Figure C2 in Appendix C showing the spatial and temporal distribution of POC, similar to Figures 1 and 2, and POC violin and box plots (similar to Figure 5). A short description will also be added to the end of Sections 3.1 and 3.3 referencing these figures.

For the dataset, the ID is very strange. You have something like "AC, AF", you also have something as river names, numbers ("113006A"). Can you set a rule for your ID.

As OzRiCa compiles data from multiple sources, including government databases, literature, and our own field campaigns, we prefer to retain the original site IDs. This is important for traceability, as many IDs (e.g. "113006A") correspond to established monitoring stations within state databases. Keeping these original site IDs will allow users to link each record back to its original source. To improve clarity, we have added a 'River.Name' column to data tables A and B, which reports the stream or river name for all sites where this information is available (i.e. > 5,300 rows).

Additionally, it can be very easy for you to get the catchment information for each site. Can you link your data to global hydrograph dataset or national hydrograph datasets, such as Hydrosheds?

Thank you for this suggestion. We did consider linking OzRiCa to national or global hydrographic datasets such as HydroSHEDS. However, to do so accurately, each site must first be associated with a delineated catchment. From our own experience delineating all sites we sampled directly, this process involves much more than simply overlaying a DEM, it requires multiple QA/QC steps to ensure the correct catchment outlet and boundary, particularly for small or low-relief streams. To maintain data integrity, we chose not to include catchment information unless it could be verified with high confidence as we did with the sites from our sampling campaigns.